# Pulmonary Edema in COVID-19 Treated with Furosemide and Negative Fluid Balance (NEGBAL): A Different and Promising Approach

**DOI:** 10.3390/jcm10235599

**Published:** 2021-11-28

**Authors:** Jose L. Francisco Santos, Patricio Zanardi, Veronica Alo, Marcelo Rodriguez, Federico Magdaleno, Virginia De Langhe, Vanina Dos Santos, Giuliana Murialdo, Andrea Villoldo, Micaela Coria, Diego Quiros, Claudio Milicchio, Eduardo Garcia Saiz

**Affiliations:** 1Intensive Care Unit, Clinica Colon, Mar del Plata, Buenos Aires 7600, Argentina; patriciozanardi@gmail.com (P.Z.); veronica_alo@hotmail.com (V.A.); vaninadossantos@hotmail.com (V.D.S.); giulieme38@hotmail.com (G.M.); andrevilloldo@gmail.com (A.V.); mikcoria@gmail.com (M.C.); 2Cardiology Service, Clinica Colon, Mar del Plata, Buenos Aires 7600, Argentina; marceloarodriguez@gmail.com; 3Diagnostic Imaging Service, Hospital Privado del Sur. Bahía Blanca, Buenos Aires 8000, Argentina; fedemagdaleno@hotmail.com (F.M.); virgi_dl@hotmail.com (V.D.L.); 4Diagnostic Imaging Service, Clinica Colon, Mar del Plata, Buenos Aires 7600, Argentina; dquiros@copetel.com.ar (D.Q.); claudiomilicchio@yahoo.com.ar (C.M.); egarciasaiz@yahoo.com.ar (E.G.S.)

**Keywords:** COVID-19, furosemide, edema, volume overload, diuretic, NEGBAL

## Abstract

In COVID-19, pulmonary edema has been attributed to “cytokine storm”. However, it is known that SARS-CoV2 promotes angiotensin-converting enzyme 2 deficit, increases angiotensin II, and this triggers volume overload. Our report is based on COVID-19 patients with tomographic evidence of pulmonary edema and volume overload to whom established a standard treatment with diuretic (furosemide) guided by objectives: Negative Fluid Balance (NEGBAL approach). Retrospective observational study. We reviewed data from medical records: demographic, clinical, laboratory, blood gas, and chest tomography (CT) before and while undergoing NEGBAL, from 20 critically ill patients. Once the NEGBAL strategy was started, no patient required mechanical ventilation. All cases reverted to respiratory failure with NEGBAL, but subsequently two patients died from sepsis and acute myocardial infarction (AMI). The regressive analysis between PaO2/FiO2BAL and NEGBAL demonstrated correlation (*p* < 0.032). The results comparing the Pao2Fio2 between admission to NEGBAL to NEGBAL day 4, were statistically significant (*p* < 0.001). We noted between admission to NEGBAL and day 4 improvement in CT score (*p* < 0.001), decrease in the superior vena cava diameter (*p* < 0.001) and the decrease of cardiac axis (*p* < 0.001). Though our study has several limitations, we believe the promising results encourage further investigation of this different pathophysiological approach.

## 1. Introduction

In December 2019, a new coronavirus, known as severe acute respiratory syndrome coronavirus 2 (SARS-CoV-2), emerged in Wuhan, China [1] and spread throughout the world [2,3].

In COVID-19, pulmonary edema is described [4,5,6,7]; however, the dominant paradigm is focused on cytokine storm [8,9,10,11,12] as responsible for lung injury and subsequent acute respiratory distress syndrome (ARDS) [5,13]. Not everyone agreed with this paradigm. Sinha et al. [14] challenged the role of this cytokine storm given that median IL-6 levels in non-COVID patients ARDS are up to 200 times higher than in patients with severe COVID-19. Gattinoni et al. [15] maintained that COVID-19 presented as an “atypical form” of ARDS.

On the other hand, Kuba et al. and Imai et al. reported that angiotensin-converting enzyme 2 (ECA2) levels during a SARS-CoV infection are decreased [16,17]. Furthermore, in patients with COVID-19, plasma levels of Angiotensin II are higher than in healthy population [18] and stimulate an upregulation of aldosterone level, triggering sodium and water retention [19,20,21,22]. SARS-CoV-2 enters through ACE2, and further downregulates ACE2 [18,23,24] expression so that this enzyme is unable to exert its protective effects. The dysregulated activity of the renin angiotensin aldosterone system (RAAS) [16] is partly responsible for pulmonary edema in COVID-19 [16,24].

ACE2 is known for its effect as the main counter-regulatory mechanism for the renin–angiotensin aldosterone system (RAAS), which is an essential player in blood pressure control by retaining sodium and water and increasing the intravascular fluid volume. SARS-CoV-2 binds ACE2 and accelerates the degradation of ACE2, and thus decreases the counteraction of ACE2 on RAAS. The final effect is increasing reabsorption of sodium and water, and therefore causing volume overload [16,19,20,22].

The RAAS can be envisioned as a dual function system in which the vasoconstrictor/proliferative or vasodilator/antiproliferative actions are primarily driven by the ACE–ACE2 balance [19]. According to that, an increased ACE/ACE2 activity ratio generated by the downregulation action of SARS-CoV2 on ACE2 [18,22,23,24] will lead to increased Angiotensin II [18] and increased catabolism of Angiotensin 1–7, towards vasoconstriction, endothelial dysfunction, prothrombosis, proinflammatory, and antinatriuretic effect [19,22].

Acute pulmonary edema is caused mostly by one of the following mechanisms: pulmonary venous pressure elevation—volume overload—or augmentation of the alveolar capillary membrane permeability—inflammation [25]. In fact, both mechanisms sometimes coexist and the distinction is irrelevant.

There are bibliographic mentions of pulmonary edema in COVID 19 [4,5,6,7], as well as evidence of volume overload in COVID-19: Lang et al. describes frequent and pronounced vasculature in affected lung areas that may be suggestive of disordered vasoregulation [26]. Eslami et al. observed increased cardiothoracic ratio [27] and it is also described as right ventricular dilatation [28,29].

In this setting, a different approach emerged: moderate or severe COVID-19 could experience a severe acute pulmonary edema with a “dual hit”. A “first hit” of pneumonitis—augmentation of the alveolar capillary membrane permeability—can lead to low hydrostatic pressure pulmonary edema. The “second hit” is high pressure pulmonary edema, caused by increase of hydrostatic pressure [25,30] secondary to volume overload, a result of dysregulation of the RAAS [16,17,18,19,20,21,22]. This results in a “dual hit” that triggers severe acute pulmonary edema.

If this edema does not resolve, then comes a “third hit” with secondary inflammation, superinfection, fibrosis, and finally the typical ARDS. With all of this in mind, we looked for and detected pulmonary edema before ARDS was triggered.

We searched cases of moderate and severe COVID-19, and found tomographic evidence of pulmonary edema, detected by dilated superior vena cava, large pulmonary arteries, diffuse interstitial infiltrates, and dilated right ventricle [31].

At the detection of pulmonary edema in COVID-19 patients, we established a standard treatment consisting of oral hydric restriction and diuretics [25,32]. The effects of furosemide on pulmonary edema were well established decades ago [25].

At the time of submission of our study, we did not find any literature that proposed the model of pulmonary edema and volume overload secondary to the dysregulation of the renin angiotensin aldosterone system in COVID-19.

## 2. Materials and Methods

### 2.1. Ethical Approval

The Ethics in Investigation Committee (Mar del Plata, Argentina) approved the study. All investigations were conducted in accordance with the 1964 Declaration of Helsinki and its later amendments.

Patients were not involved in the design, or conduct, or reporting, or dissemination plans of our research.

### 2.2. Study Design and Patient Population

This single-center retrospective, observational study was conducted on patients from 22 June 2021 to 16 August 2021, with confirmed diagnosis of COVID-19 and pulmonary edema, who were admitted to our high complexity center in Mar del Plata, Argentina, and underwent a treatment with furosemide in continuous intravenous infusion, guided by objectives: Negative Fluid Balance approach (NEGBAL).

Inclusion criteria were as follows: (1) confirmed diagnosis of COVID-19 through real-time reverse transcriptase polymerase chain reaction (RT-PCR) assay with samples obtained from nasopharyngeal swab or positive antinucleocapsid IgM antibodies; (2) PaO2/FiO2 (ratio of arterial oxygen partial pressure to fractional inspired oxygen) <200; (3) age older than 18 years, and (4) tomographic evidence of acute pulmonary edema, defined as dilated superior vena cava, large pulmonary arteries, diffuse interstitial infiltrates with Kerley lines, and dilated right ventricle or dilated cardiac axis.

Exclusion criteria were as follows: (1) patients with prior indication for diuretics for another reason, (2) renal failure, (3) cardiac failure (diagnosis by echocardiography), (4) hepatic failure, (5) hypernatremia or hyponatremia, (6) hypotension or shock.

We reviewed data from medical records of 20 consecutive adult patients: demographic; clinical, laboratory; Pro b-type natriuretic peptide (pro-BNP, negative: below 125 pg/mL); high-sensitivity cardiac troponin (hs-cTnT, negative: <14 ng/L); blood gas; chest tomography (CT); the oxygen therapy support; and mechanical ventilation (MV) requirements, all of which were reviewed and recorded by investigators. With the purpose of knowing the patient’s basal hematocrit out of the course of COVID-19, we reviewed prior hematocrit, if any, defined as hematocrit previous to COVID 19 infection (hematocrit prior to admission to NEGBAL).

The treatments for COVID-19 pneumonia, in our series, were based on standard recommendations. As concomitant interventions these patients received dexamethasone 6 mg/day and thromboembolic prevention with enoxaparin 40 mg/day.

### 2.3. Chest CT Imaging

CTs were performed upon admission to NEGBAL and CT controls were scheduled for day 4 (+/− 1 day), day 8 (+/− 1 day), and day 12 (+/− 1 day). We also reviewed tomographies performed before admission to our center, at the emergency room or as an inpatient, if any, during the first days of the course of COVID-19 and distanced at least two days before admission to NEGBAL (CT score prior to NEGBAL) with the purpose of observing the natural evolutionary trend of COVID-19 before NEGBAL.

CT was performed in every patient using a 32-slice scanner (Siemens somatom scope, Germany) or a 16-slice scanner (General Electric brivo, GE Medical System) in supine position during end-inspiration. A 1.5 mm slice thickness and 1.5 mm interval were used for the axial image.

For the evaluation of CT infiltrates, the score described by Pan F et al. [33] was used as a semiquantitative measurement, measuring the sum of lung involvement—5 lobes—each lobe, on a scale from 0 (normal) to 5 (maximum infiltrate), with a maximum CT score of 25. The measurement of the superior vena cava diameter (Ø svc) in the CT was done just above the arch of the azygos veins. The cardiac axis (Ø card.) was also measured: transverse measurement across the four cavities. Cardiothoracic ratio (CTR) was defined as the greatest transverse cardiac diameter from outer to outer myocardium divided by the greatest transverse thoracic diameter from inner to inner chest wall. These determinations were calculated and reviewed by two external imaging specialists. Both radiologists were blinded to laboratory data, clinical features, and patients’ diagnosis.

### 2.4. Negative Fluid Balance (NEGBAL) Approach

At the tomographic detection of pulmonary edema (dilated superior vena cava, large pulmonary arteries, diffuse interstitial infiltrates with Kerley lines, and dilated right ventricle or dilated cardiac axis), we established treatment: NEGBAL approach. It consisted of oral hydric restriction and diuretics (20 mg of furosemide, intravenous bolus, followed by furosemide in endovenous continuous infusion, starting at 60 mg/d). The objective was to achieve negative fluid balance, between 600 to 1400 mL/d adjusted to body surface area, with a final target of 8–10% of body weight in 8 days. The furosemide dose was titrated considering heart rate and blood pressure, target fluid balance, hematocrit, and serum creatinine. The presence of hypotension (systolic blood pressure less than 100 mmHg for 30 min), hyponatremia, hypernatremia, or elevated serum creatinine was considered a cause of suspension of NEGBAL. All patients were followed until either death or complete recovery and discharge were reached.

### 2.5. Statistical Analysis

To analyze the relationship between the variation of the PaO2/FiO2 ratio and the NEGBAL variables, we used a linear regression model of the form PAFIBAL~β0+β1NEGBAL. For the model, the response variable PAFIBAL was registered as the difference between the PaO2/FiO2 at admission to NEGBAL and the PaO2/FiO2 at day 7. Continuous variables were expressed as means. Categorical variables were summarized as counts. No changes (adjustments) were made for missing data. The paired sample T test was used. All statistical tests were 2-tailed. A *p*-value < 0.05 was considered statistically significant. The analysis has not been adjusted for multiple comparisons and, given the possibility of a type I error, the findings should be interpreted as exploratory and descriptive. All analyses were performed using R software, version 4.1.1 (R Foundation for Statistical Computing) (see Appendix A).

## 3. Results

Between 22 June and 16 August 2021, a total of 69 patients with COVID-19 were referred to our center. Forty-four patients with PaO2/FiO2 > 200 and 5 with kidney or heart failure were excluded from the analysis. Thus, data from 20 critically ill patients with laboratory-confirmed COVID-19 were analyzed. Mean age of patients was 52.9 ± 14 years old, 13 were male. The average APACHE II score at admission was 9.05 (±4.32). Among the reported population, 14 out of 20 had comorbidities and 11/20 received the first dose of vaccine. The baseline predictors of poor prognosis were 6/20 patients with obesity (IMC > 30), 4/20 patients with hypertension, 2/20 with COPD, and there were no cases of asthma or pulmonary fibrosis, 6/19 patients had elevated D-dimers on admission. (See demographic information in Table 1 and biochemical data at admission information in Table 2).

In our series of patients, the requirement of oxygen therapy support was: 2/20 required MV, 3/20 required high flow nasal cannula, 2/20 patients required noninvasive ventilation, 10/20 required oxygen mask plus reservoir, and 3/20 standard oxygen mask.

At the time of starting NEGBAL, 2/20 (cases 1 and 2) were already under MV. After the establishment of the NEGBAL approach, no patient required MV. Cases 1 and 2 improved oxygenation and CT score with NEGBAL, and successful extubation were achieved in both cases. Two patients died, case 1, six days after ending NEGBAL due to sepsis and case 17, five days after ending NEGBAL, due to acute myocardial infarction. The remaining 18 patients were discharged.

The mean total accumulated negative fluid balance was −7637 mL (±2616 mL). Adjusted for total days of effective application of NEGBAL in the population, the mean negative fluid balance obtained was −1184 mL for each day. Adjusted to square meter, the mean negative fluid balance was −3952 mL/m^2^ (Appendix A).

In the population studied, the heart rate stayed within normal range, with a tendency to bradycardia and they remained afebrile (Appendix A).

The safety data showed that none of the 20 cases presented electrolyte or serum creatinine alterations or hypotension during NEGBAL. For this reason, none of the patients presented criteria for suspending NEGBAL (Appendix A).

Correlational analysis was performed between PaO2/FiO2-NEGBAL and NEGBAL accumulated; the results were statistically significant (*p* = 0.034) (Figure 1A).

PaO2/FiO2 at admission to NEGBAL (mean 118 ± 47) was compared with PaO2/FiO2 on day 4 (mean 246 ± 111); the differences was statistically significant (*p* < 0.001; (95% CI, −168, −89)), on day 8 (mean 316 ± 90) (*p* < 0.001; (95% CI, −244, −157)), and at discharge (mean 332 ± 80) (*p* < 0.001; (95% CI, −256, −172)). A significant improvement in oxygenation was achieved in all cases (Figure 1B).

The CT score was analyzed and the difference was statistically significant. CT score prior to NEGBAL (mean 7.1 ± 1.4) was compared with CT score at admission (mean 17.2 ± 1.3) (*p* < 0.001; (95% CI, −12.1, −8.2)), CT score at admission (mean 17.2 ± 1.3) to day 4 (mean 10.6 ± 1.6) (*p* < 0.001; (95% CI, 4.1, 8.9)) and CT score day 4 (mean 10.6 ± 1.6) to CT score day 8 (mean 7.7 ± 1.4) of NEGBAL (*p* < 0.001; (95% CI, 1.4, 4.3)) (Figure 2A).

The hematocrit level before starting NEGBAL (mean 41.5 ± 5.6%) was compared to the hematocrit upon admission to NEGBAL (mean 37.5 ± 4.9%) (*p* < 0.001; (95% CI, 3.7, 7.6)) and the admission hematocrit (mean 37.5 ± 4.9%) to the hematocrit at discharge (mean 40.7 ± 5%) (*p* = 0.002; (95% CI, −5.1, −1.2)) (Figure 2B).

The diameters of the superior vena cava were measured in millimeters and compared, before starting NEGBAL (mean 14.6 ± 4.4) to admission to NEGBAL (mean 17.9 ± 3.8) (*p* < 0.001; (95% CI, −6.6, −2.4)), admission to NEGBAL (mean 17.9 ± 3.8) to NEGBAL day 4 (mean 14.2 ± 4.5) (*p* < 0.001; (95% CI, 1.7, 5.5)) and admission to NEGBAL (mean 17.9 ± 3.8) to discharge (mean 13.9 ± 3) (*p* < 0.001; (95% CI, 2.4, 5.5)) (Figure 2C).

The cardiac axis was measured in centimeters and compared, prior to NEGBAL (mean 12.2 ± 1.4) to admission to NEGBAL (mean 12.8 ± 1.3) (*p* = 0.002; (95% CI, −1.6, −0.4)), admission to NEGBAL (mean 12.8 ± 1.3) to NEGBAL day 4 (mean 11.7 ± 1.6) (*p* < 0.001; (95% CI, 0.6, 1.5)) (Figure 2D). More information: Table 3 and Appendix A.

The average dose of intravenous infused furosemide in milligrams per hour was: day 1, 3.5 (±2); day 2, 4.2 (±2); day 3, 4.5 (±3); day 4, 4.5 (±3); day 5, 4.8 (±3); day 6, 4.8 (±4); and day 7, 5.5 (±4) (Appendix A).

## 4. Discussion

The present study has several limitations. It is retrospective, observational, and the number of patients quite limited. Added to the absence of a control group, it is that these limitations could affect the interpretation of some result. 

We observed in our series an incidence of obesity and hypertension similar to that published by Bonifazi et al. [34], although with a lower incidence of smokers and no cases of asthma. Furthermore, in the biomedical published literature, unfortunately, we did not find a similar approach for pulmonary edema in COVID-19. However, there are numerous reports of pulmonary edema [4,5,6,7,16,17] together with evidence of volume overload [26,27,28,29] in COVID-19. This evidence, added to the promising clinical response to NEGBAL, support this approach to pulmonary edema in COVID-19 as a biological plausibility.

It is encouraging that the population in this report obtained a significant improvement in oxygenation between PaO2/FiO2 at the beginning of NEGBAL and PaO2/FiO2 at day 4 (*p* < 0.001). An encouraging point is that no patient with moderate and severe COVID-19 required MV while implementing this NEGBAL approach.

We were able to observe COVID-19 progression as the hematocrit level decreased and increases in the CT score in the period elapsed between an evaluation for COVID-19 days before and admission to NEGBAL. This deterioration shows how COVID-19 progresses gradually with volume overload and worsening of pulmonary edema evidenced in a higher CT score and a dilutional decrease in hematocrit. Interestingly, when NEGBAL was started, the decrease in hematocrit and tomographic deterioration was reversed.

We looked for indirect evidence that could reflect the existence of volume overload, probably generated by RAAS deregulation in COVID-19, and provide the following observations:

First, the variability of the diameter of the superior vena cava, which increased as COVID-19 worsened (*p* < 0.001), could be a sign of an increase of fluid in the vascular compartment. Furthermore, when NEGBAL started, we detected a progressive and significant (*p* < 0.001) decrease in this diameter. The dilation of the cavities and the vena cava were mentioned by other authors and, similar to our observations, were associated with the severity of COVID-19 [29]. Lang et al. [26] also describes a pronounced dilatation of pulmonary vasculature. We interpreted that this description may be suggestive of volume overload.

Second, we observed that the length of the cardiac axis showed a similar behavior. There was a progressive increase as COVID-19 worsened (*p* = 0.002), but as NEGBAL was implemented, we detected a gradual decrease in the cardiac axis (*p* < 0.001). We observed the same behavior in the cardiothoracic index. This coincides with Eslami et al. [27], who revealed that increased cardiothoracic ratio (CTR) is a powerful predictor of mortality. However, the authors could not associate the increased cardiac index with markers of heart disease. This supports our hypothesis that the observed cardiovascular dilation is due to a state of hypervolemia and not to cardiac dysfunction.

Third, anemia in COVID-19 has numerous explanations [35,36,37], but none have been conclusive. In our series, the behavior of the hematocrit in COVID-19 patients manifested itself in coherence with the observations outlined earlier. We observed a progressive decrease of hematocrit as COVID-19 worsened (*p* = <0.001), interpreting it secondary to a dilution effect. As NEGBAL was being implemented, it was noticeable that the hematocrit showed a progressive rise (*p* = 0.002). Tao et al. [35] described that patients with severe anemia presented a higher proportion of hypoxia than patients with mild anemia. This is consistent with our observation that, as COVID-19 progresses, there is more anemia (for hemodilution) and more hypoxemia (for edema), and when NEGBAL was implemented, and abnormal hypervolemia corrected, hypoxemia improved and hematocrit increased.

A particular pattern was repeated in all cases as COVID-19 worsened. There was a decrease in hematocrit level, coinciding with an increase in the diameter of the superior vena cava. These two simultaneous observations, we believe, can only be explained by a state of hypervolemia secondary to excess of fluids.

Fourth, we observed that heart rate showed a tendency to bradycardia. Sinus bradycardia is common and paradoxical in COVID-19 [38,39,40] and have been related to favipiravir [41], SIADH [42], hydroxychloroquine [43], remdesivir [44,45], lopinavir–ritonavir [46], among other multiple explanations. It is interpreted that it may be another indirect evidence of abnormal hypervolemia, which generates an increase in preload and, by Frank Starling’s law, increase in stroke volume, resulting in a cardiac adaptive response, with a decrease in heart rate.

Fifth, the rapid improvement in the CT score within day 4 (*p* < 0.001) to day 8 (*p* < 0.001) after NEGBAL, allowed us to suspect that would predominate as the cause of these infiltrates is pulmonary edema (secondary to “dual hit”) and not hegemonic inflammation.

Sixth, we observe a correlation with statistical significance between NEGBAL and PaO2/FiO2 (*p* = 0.034), which supports the hypothesis that in COVID-19 there is pulmonary edema and there is no typical ARDS. However, this “dual hit”, inflammation plus edema, could be part of a process that would end, at a later stage, in ARDS [47,48].

Seventh, and finally, the mean accumulated negative fluid balance was −7637 mL, with optimal hemodynamic tolerance in all patients. We believe that adequate tolerance was only possible due to the pre-existence of volume overload.

These observations could support our proposed explanation of successive pulmonary hits.

The “first hit” of COVID-19 pneumonitis [49], followed by dysregulation RAAS [17] “second hit”, generating excess of intra- and extravascular fluid, causing volume overload [26,28,29] and pulmonary edema [4,5,6,7]. With unresolved edema, appeared a “third hit” characterized by infection, fibrosis, more inflammation, and ending in ARDS [47,50] (see Appendix A).

For the aforementioned, we believe that as volume overload increases, there are more dilated distal subpleural vessels [26], more dilatation of the large blood vessels [27,28,29]. A posteriori, it ensues a dilatation of the superior vena cava, progressively, also the cardiac index [27] is added, and the cardiac axis is expanded, increasing the dilatation of the right ventricle [28,29]. This causes an increase of the hydrostatic pressure [25,30] and the consequent pulmonary edema [4,5,6,7,16,17]. Once the negative fluid balance approach was established, this harmful sequence was reverted.

We believe that dosing angiotensin II, angiotensin 1–7, vasopressin and aldosterone daily from the beginning of the COVID-19 infection could detect a dysregulation of the RAAS [51,52,53,54].

To detect and quantify fluid overload in COVID-19 is challenging, and consequently interesting that Mei et al. [55] apply the remote dielectric sensing (ReDS) technique, which is a noninvasive electromagnetic wave technology that provides an accurate reading of lung fluid content, and they detected fluid overload in the lung parenchyma in COVID-19. Similarly, Rasch et al. [6] used an invasive technique of transpulmonary thermodilution that provides bedside measurement of extravascular lung water index, which is a marker of pulmonary edema. They reported that COVID-19 pneumonia has up to five times more extravascular lung water index (2600 mL) than normal lungs (500 mL). We consider that both studies support our hypothesis.

Finally, in our series, we observed the absence of clinical signs of systemic inflammatory response syndrome and discrete alterations in leukocyte count and c-reactive protein. This supports the Sinha et al. [14] hypothesis that the cytokine storm would be just a discrete drizzle.

## 5. Conclusions

Although it is a study with several limitations, this pathophysiological approach of pulmonary edema and volume overload due to downregulation ACE2 and pneumonitis plus edema, “dual hit” shows a promising perspective. We are hopeful that these findings will encourage research towards this pathophysiological and therapeutic approach in COVID-19. Prospective, randomized, and multicenter case-control studies are necessary to determine whether this different pathophysiological perspective has a real basis or if it is a mere coincidence for an attractive but failed hypothesis.

## Figures and Tables

**Figure 1 jcm-10-05599-f001:**
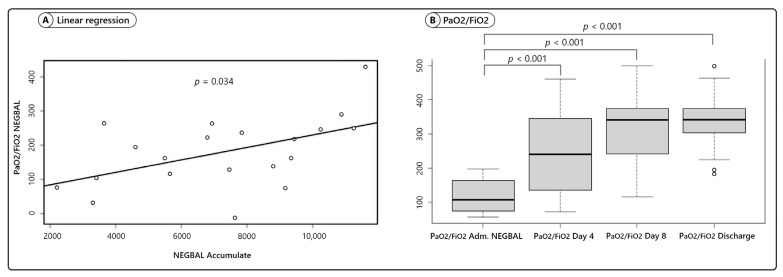
PAFI evolution during NEGBAL approach. (**A**) Scatter plot and regression line for PAFIBAL onto NEGBAL. (**B**) Boxplot for the PaO2/FiO2 ratio on the day of admission, day 4, day 8, and on day of discharge.

**Figure 2 jcm-10-05599-f002:**
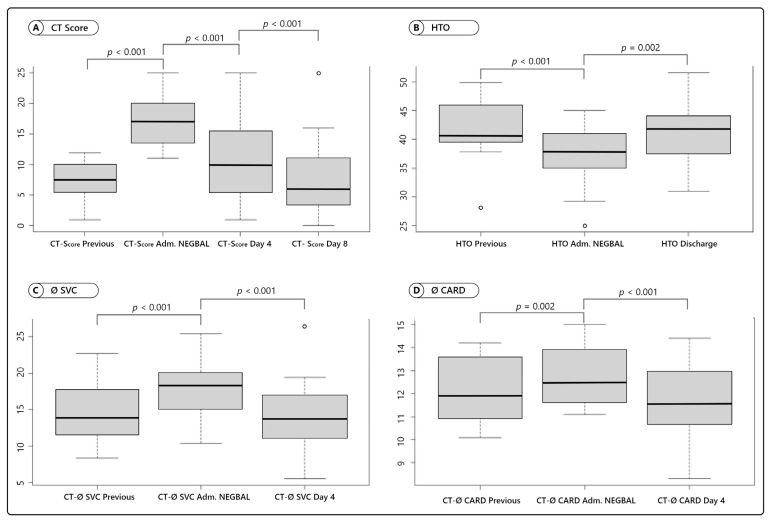
CT score evolution pre- and post-NEGBAL of CT, HTO, Ø SVC, and Ø CARD. (**A**) Box plot for CT score previous to NEGBAL, CT score adm. NEGBAL, CT day 4, and CT day 8. (**B**) Box plot for previous HTO, HTO adm. NEGBAL, and HTO day 4. (**C**) Box plot for Ø SVC previous NEGBAL, Ø SVC adm. NEGBAL, and Ø SVC day 4. (**D**) Box plot for Ø CARD previous NEGBAL, Ø CARD adm. NEGBAL, and Ø CARD day 4.

**Table 1 jcm-10-05599-t001:** Demographic data and background. Hematocrit, Pao2Fio2, and CT score at admission of the 20 cases.

CASE	APACHE II	VAC-1	VAC-2	BMI	DBT	CARD	SMK	HCT	PaO2/FiO2	CT Score
1	21	SPK		27		HT	No	43.1	60	21
2	13			23			No	25	62	13
3	13	A-Z		28			No	37	65	20
4	8	SNP		24			No	29.3	188	17
5	3			33			No	42.7	56	17
6	7			27			No	43	110	14
7	13	SPK		25		AF	No	35	61	23
8	9	SPK		35			Yes	41	100	19
9	7			32			No	38	177	18
10	9	A-Z		26	DBT2		No	37.4	86	25
11	3			33			No	38	120	12
12	10	SNP		31			No	35.3	121	20
13	9	SNP	SNP	25		HT	No	45	101	16
14	8	SNP		27			Yes	39	185	11
15	5			28		HT	No	35.7	145	22
16	3			22			No	41	197	12
17	12			22	DBT2	HT	No	38.3	102	17
18	10	SPK		25		AF	Yes	34.9	160	17
19	7	A-Z		25			No	31.3	104	11
20	5			33	DBT2		No	40	162	19

Abbreviations: VAC-1: Vaccine first dose; VAC-2: Vaccine second dose; BMI: Body Mass Index; DBT: Diabetes CARD: Cardiovascular History; SMK: Smoking; HCT: Hematocrit at admission; SPK: Sputnik; A-Z: AstraZeneca; SNP: Sinopharm; HT: Arterial Hypertension; AF: Atrial fibrillation.

**Table 2 jcm-10-05599-t002:** Biochemical data at admission.

CASE	COVID-19 Diagnosis	HCT	LEUK	LYMP	D-Dimer	proBNP	Troponin	CRP	PCT
1	PCR+	43.1	12,140	364	NO	1191	10		0.2
2	PCR+	25	16,760	1710	0.04		8		0.38
3	PCR+	37	22,000	1180	0.2	655	6		
4	PCR+	29.3	8450	2130	0.33	253	7	48.9	0
5	PCR+	42.7	9210	1234	0.37	16	11	15.5	0.1
6	PCR+	43	16,560	894	0.92	35	3	16.6	
7	PCR+	35	16,830	539	0.19				
8	PCR+	41	8000	770	2.4	330	31		
9	PCR+	38	8510	885	1.09	280	6	18	0.07
10	PCR+	37.4	14,450	795	0.55	109			0.8
11	PCR+	38	5850	1433	0.19	133	5	33	0.05
12	PCR+	35.3	13,670	1121	0.48	351	5	44.9	0
13	PCR+	45	17,700	779	0.22	80	9	40	0.17
14	PCR+	39	16,900	3700	0.25	10	4	18	0
15	PCR+	35.7	14,090	1043	0.91	111	6	52.6	0.31
16	IgM+	41	6900	1553	0.19	18	7	15.8	0.05
17	PCR+	38.3	7050	455	>4.4	367	6	35	
18	PCR+	34.9	16,500	446	<0.19	537	5	37	0.24
19	PCR+	31.3	6530	581	0.45	NO	4	59.4	0.05
20	PCR+	40	6660	733	0.26	85	5	46	0.08

Abbreviations: PCR: Polymerase Chain Reaction; IgM: Anti-nucleocapsid IgM antibodies; HCT: Hematocrit; LEUK: Leukocytes; LYMP: Lymphocytes; proBNP: pro-B-type Natriuretic Peptides; CRP: C-Reactive Protein; PCT: Procalcitonin.

**Table 3 jcm-10-05599-t003:** Statistical analysis of PaO2FiO2, CT score, diameter SVC, cardiac axis, CTR, and hematocrit.

Variable	Prior NEGBAL			Admission NEGBAL			Day 4 NEGBAL	
	Mean	SD	*p* Value ^(a)^	Mean	SD	*p* Value ^(b)^	Mean	SD
PaO2FiO2				118	±47	*p* < 0.001	246	±111
CT Score	7.1	±1.4	*p* < 0.001	17.2	±1.3	*p* < 0.001	10.6	±1.6
Diameter SVC, mm	14.6	±4.4	*p* < 0.001	17.9	±3.8	*p* < 0.001	14.2	±4.5
Cardiac Axis, cm	12.2	±1.4	*p* < 0.001	12.8	±1.3	*p* < 0.001	11.7	±1.6
CTR	0.45	±0.05	*p* = 0.002	0.49	±0.04	*p* < 0.001	0.45	±0.06
Hematocrit, %	41.5	±5.6	*p* < 0.001	37.5	±4.9	*p* = 0.002	Discharge Day	
							40.7	±5

^(a)^ This column presents *p* value for comparisons between prior NEGBAL values and admission NEGBAL values. ^(b)^ This column presents *p* value for comparisons between admission NEGBAL values and day 4 NEGBAL values or at discharge (only for hematocrit). Abbreviations: SD: Standard Deviation; SVC: Superior Vena Cava; CTR: Cardiothoracic Ratio.

## Data Availability

The data presented in this study are available on request from the corresponding author. More data are contained within the Appendix A. The visualization of the tomographies is available in an online website.

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
