# Peer review of "Pulmonary Edema in COVID-19 Treated with Furosemide and Negative Fluid Balance (NEGBAL): A Different and Promising Approach"

_jcm, 2021, doi:10.3390/jcm10235599_

Round 1
Reviewer 1 Report
Santos J.L.F et al. reported a really interesting paper about innovative therapeutic approach for pulmonary edema in patients with COVID-19 pneumonia.
The manuscript is so elegant and has been well written, but i would like to express some minor concerns:
- Data analyses didn’t include some relevant baseline predictors of poor prognosis such as Obesity or Respiratory diseases (COPD, asthma or pulmonary fibrosis) (please see Predictors of Worse Prognosis in Young and Middle-Aged Adults Hospitalized with COVID-19 Pneumonia: A Multi-Center Italian Study (COVID-UNDER50) Clin. Med. 2021, 10, 1218. Factors associated with COVID-19-related death using OpenSAFELY. Nature; Volume 584: 20 August 2020; ); therefore i would suggest to add these factors and their potential prognostic role in terms of prognosis
- I think it might be conceivable to include a description of Oxygen therapy support (Non invasive ventilation, High flow nasal canula, Oxygen mask plus resvoir etc..) performed in all patients, except for two patients already in MV at NEGBAL starting time
- Pulmonary edema has been already established as a pathogenetic factor in COVID-19 pneumonia; lung fluid excess can be assessed by means of invasive methods or through non invasive new techniques, as recently reported (please see : “Validation of remote dielectric sensing (ReDS) in monitoring adult patients affected by COVID 19 pneumonia”DIagnostics2021; 11(6): 1003
Reviewer 2 Report
The present paper deals with effectiveness and safety of Negative Fluid Balance approach with diuretic (furosemide) in COVID-19 patients with radiological evidence of pulmonary edema and volume overload. This work is very interesting, but I have some comments:
- This is a retrospective observational study on 20 patients and there was no a control group, These major limitations, particularly the absence of a control group, are likely to impact results interpretation, and should be at least mentioned in the discussion. Evolution over time of volume overload has been assessed in a previous study from Mei F. et al (Validation of Remote Dielectric Sensing (ReDS) in Monitoring Adult Patients Affected by COVID-19 Pneumonia.Diagnostics (Basel). 2021), using an innovative tool (Remote Dielectric Sensing -ReDS), showing a significant progressive clinical improvement and edema reduction in 14 days with “standard” therapy. Although study population and tools used for measuring fluid overload are different, the authors should acknowledge the possibility that edema resolution could revert also thank to the standard treatment, as the absence of a control group does not allow to declare that NEGBAL was the only effective drug in improving outcomes.
- In the abstract, the author declared that “All cases reverted to respiratory stable condition”, but in the main text, it is written that two deaths occurred. Please clarify.
- The significant significant improvement in oxygenation between PaO2/FiO2 at the beginning of NEGBAL and PaO2/FiO2 at day 4 (p<0.001) might be due also to other factors, including associations with other medications (i.e. steroids, anticoagulants etc). There is no mention of dose and type of “standard” treatments for COVID pneumonia, but it would be important to describe all concomitant interventions these patients received.
- supplementary materials are redundant and should be shortened
Round 2
Reviewer 2 Report
The authors properly answered